# *Bacillus*-Based Biocontrol Agents Mediate Pathogen Killing by Biodegradable Antimicrobials from Macrolactin Family

**DOI:** 10.3390/ijms262211167

**Published:** 2025-11-19

**Authors:** Elena B. Guglya, Olga A. Belozerova, Anton E. Shikov, Vera A. Alferova, Maria N. Romanenko, Vladimir K. Chebotar, Maria S. Gancheva, Maria E. Baganova, Ekaterina A. Vinogradova, Elizaveta A. Marenkova, Vladislav A. Lushpa, Anna A. Baranova, Margarita N. Baranova, Olga A. Shevtsova, Arsen M. Kudzhaev, Yuri A. Prokopenko, Sergey I. Kovalchuk, Dmitrii A. Lukianov, Kirill S. Antonets, Anton A. Nizhnikov, Stanislav S. Terekhov

**Affiliations:** 1Shemyakin-Ovchinnikov Institute of Bioorganic Chemistry, Miklukho-Maklaya 16/10, 117997 Moscow, Russia; eguglya@gmail.com (E.B.G.); o.belozyorova@gmail.com (O.A.B.); alferovava@gmail.com (V.A.A.); kategrape@gmail.com (E.A.V.); marenkova.lizzz@gmail.com (E.A.M.); lushpa1696@gmail.com (V.A.L.); anjabaranowa@list.ru (A.A.B.); baranova@ibch.ru (M.N.B.); kudzhaev_arsen@mail.ru (A.M.K.); tetrahydrofuran@mail.ru (Y.A.P.); xerx222@gmail.com (S.I.K.); 2Biological Faculty, St. Petersburg State University, 199034 St. Petersburg, Russia; a.shikov@arriam.ru (A.E.S.); m.romanenko@arriam.ru (M.N.R.); m.gancheva@spbu.ru (M.S.G.); k.antonets@arriam.ru (K.S.A.); a.nizhnikov@arriam.ru (A.A.N.); 3All-Russia Research Institute for Agricultural Microbiology, 196608 St. Petersburg, Russia; vladchebotar@arriam.ru (V.K.C.); mashul991@mail.ru (M.E.B.); 4Moscow Center for Advanced Studies, Kulakova Str. 20, 123592 Moscow, Russia; 5Department of Chemistry, Lomonosov Moscow State University, Leninskie Gory, 119992 Moscow, Russia; shevtsovaolga7@yandex.ru (O.A.S.); dmitrii.a.lukianov@gmail.com (D.A.L.); 6Center for Molecular and Cellular Biology, 121205 Moscow, Russia

**Keywords:** biocontrol agents, *Bacillus velezensis*, macrolactins, biotransformation, antibiotic resistance, activity-guided metabolomics

## Abstract

The transition to organic farming is one of the most desirable achievements of our time. Rational use of organic farming approaches not only enables a reduction in costs and increased yields but also limits the risks associated with the use of pesticides and chemicals. Despite the widest practical application of numerous biocontrol agents based on *Bacillus* strains, their metabolome, including the main active substances, often remains unknown. In order to understand the basic principles of the functioning of the *Bacillus velezensis* K-3618 strain, widely used in organic farming, we studied its spectrum of antimicrobial metabolites in detail. It was shown that the main antimicrobial agents of *B. velezensis* K-3618 are representatives of the macrolactin family. The identified macrolactin A (MLN A) and its acylated analogs 7-O-malonyl macrolactin A (mal-MLN A) and 7-O-succinyl macrolactin A (suc-MLN A) are active against Gram-positive bacterial pathogens, including multidrug-resistant strains. Among them, suc-MLN A is the most potent antimicrobial, highly active (MIC = 0.1 μg/mL) against the common human pathogen methicillin-resistant *Staphylococcus aureus* (MRSA). It was revealed that the primary mechanism of action of MLN A-based macrolactins is protein translation inhibition. Acylated macrolactins outperform MLN A in the prokaryotic cell-free system, displaying high efficiency in low micromolar concentrations. We observed that acylated MLN A analogs undergo pathogen-mediated biotransformation into MLN F analogs, having their antimicrobial activity reduced by two orders of magnitude. Hence, both acylation of MLNs and stabilization of the MLN A core are essential for the creation of new synthetic MLNs with improved antimicrobial activity and stability. However, we speculate that these degradability modes are of prime importance for bacterial ecology, and they are highly conserved in *Bacillus* species from various ecological niches.

## 1. Introduction

Phytopathogens pose a threat to the global economy and food industry, necessitating the search for effective counteraction measures. Application of organic farming practices and biocontrol agents enables a reduction in pesticide usage that not only manages health risks of pesticide exposure but also provides the opportunity to gain higher yields [1]. Species of the genus *Bacillus* are widely used as biological control agents acting through a synergy of mechanisms, including competition, antagonism, induction of systemic resistance, and stimulation of plant growth [2,3]. *Bacillus* species are potential producers of antibiotics that protect plants by suppressing soil phytopathogens. While a high diversity of bactericidal secondary metabolites of various chemical classes is known for *Bacillus* [4], the exact determination of the principal-acting antimicrobials in biocontrol agents is of high interest since it enables a more comprehensive and targeted practical application of these strains [5,6].

However, for the overwhelming majority of industrial biocontrol strains, the exact composition of principal-acting secondary metabolites has not been characterized in detail. *Bacillus velezensis* strains are well known for their biocontrol activity [7,8], which shields plants from a wide range of bacterial and fungal diseases and provides growth benefits to tomato, pepper, pumpkin, and cucumber [3].

To uncover the landscape of antimicrobial metabolites in a highly efficient growth-stimulating endophytic isolate, *Bacillus velezensis* K-3618 (All-Russian Collection of Agricultural Microorganisms), which is widely applied as a biocontrol agent in organic farming, we used the antimicrobial activity-guided metabolomic approach. A family of macrolactins was isolated as the main antibacterial metabolites produced by *B. velezensis* K-3618.

Macrolactins (MLNs) are macrolide natural products from marine and terrestrial microorganisms, characterized by a 24-membered lactone ring [9,10]. First reported in 1989 as MLN A–F [11], the family had grown to 33 members by 2021 and continues to expand [12]. MLNs show broad pharmacological activity, most notably antibacterial effects. Structural diversity arises from variation in the number and placement of olefinic bonds in the ring and from different post-modifications. 7-*O*-malonyl macrolactin A (mal-MLN A) and 7-*O*-succinyl macrolactin A (suc-MLN A), which carry the 7-OH position of MLN A acylated by their respective dicarboxylic acids, are produced by *Bacillus* strains, including both soil and marine species [10]. Suc-MLN was originally isolated from the fermentation broth of the marine strain *Bacillus* sp. sc026 [13], mal-MLN A was isolated later from the soil strain *B. subtilis* ICBB 1582, along with MLN A and suc-MLN A [14]. The same three compounds were isolated from *B. amyloliquefaciens* strain NJN-6, which is used for the biological control of soil-borne plant pathogens [6].

The MLN backbone is assembled by the trans-acyltransferase PKS to produce MLN A [15]. The reaction of the acylation of MLN A in the biosynthesis of mal-MLN A and suc-MLN A had remained largely enigmatic until recently, when a β-lactamase homologue, BmmI, was identified as responsible for this key step of biosynthesis [16]. BmmI can specifically attach C3–C5 alkyl acid thioesters to the 7-OH of MLN A and is substrate-promiscuous toward acyl acceptors with different backbones. The pharmacological properties of these macrolactins are diverse [9]. Preclinical studies of MLN A and suc-MLN A as antimacular and antineoplastic agents have been reported, examining their potential metabolic interactions with other drugs by assessing cytochrome P450 inhibition and induction and UDP-glucuronosyltransferase inhibition in vitro [17]. The pharmacokinetics of MLN A and suc-MLN A have been assessed as well [18,19]. MLN A and suc-MLN A have also demonstrated antifungal activity against some important plant pathogens, and the presence of the succinyl moiety at C-7 provided the compound with higher activity against all the probed fungi [20].

Macrolactins have turned out to be bacteriostatic antibiotics that inhibit a number of multidrug-resistant Gram-positive bacterial pathogens, with suc-MLN A generally exhibiting more pronounced antimicrobial activity than MLN A and mal-MLN A [14,21]. The MIC of suc-MLN A against methicillin-resistant *Staphylococcus aureus* (MRSA) has been reported as <0.25 μg/mL and 2 μg/mL against vancomycin-resistant enterococci (VRE), while for MLN A, MIC values vary from 1 μg/mL (MRSA) [21] to 10 μg/mL (MSSA) [22], with a corresponding MBC of 150 μg/mL (MSSA) [14,23].

Despite the obvious potential that macrolactins represent as antibiotics, the mechanisms of action of their numerous compounds are not fully understood. A significant contribution has been made by the recent study of the mechanism of action of MLN A [23]. The compound was shown to inhibit the first step of elongation in protein synthesis, and the bacteriostatic and/or bactericidal action was found to be species- and strain-dependent. It was demonstrated that MLN A acts as an elfamycin-type antibiotic. Various types of additional activities have also been reported for different representatives of the macrolactin family. Macrolactin N inhibits bacterial peptide deformylase with an IC_50_ value of 7.5 μM [24]. MLN S and MLN B inhibit NADPH-dependent β-ketoacyl-ACP reductase (FabG) with an IC_50_ value of 0.1 mM [25]. Transmission electron microscopy imaging has shown the impaired integrity and permeability of the cell membrane as a result of the antibacterial activity of new MLN XY against *E. faecalis* [4]. The activity of 7-*O*-succinyl macrolactin F as an inhibitor of iron uptake has also been described [26].

Prior to this study, the specific active metabolites of *B. velezensis* K-3618, as well as their mechanism of action and biodegradation pathways, were unknown. In our research, a family of macrolactins was isolated from *B. velezensis* K-3618, and the mechanism of their antibacterial activity was determined. Biotransformation of mal-MLN A and suc-MLN A was discovered, indicating the principal role of acylation and double bond architecture in their antimicrobial activity. We speculate that antimicrobial activity is strongly conserved in *B. velezensis* strains of different origins, making them promising candidates for the development of biological control agents. Further advances in semisynthetic analogs with improved stability and selectivity will provide new avenues for the development of macrolactin-based therapeutics as well.

## 2. Results

### 2.1. Cultivation of the Strain and Isolation and Identification of Metabolites with Antibiotic Activity

*B. velezensis* K-3618 exhibited prominent antibacterial activity against *S. aureus* and *E. coli* Δ*lptD* when cultivated in various liquid media (Appendix A). The major fraction of antimicrobial activity was associated with macrolactins (Figure 1A): acylated suc-MLN A, mal-MLN A, and a minor fraction of acyl-free MLN A, which was monitored using a highly sensitive alive biosensor based on the GFP-producing *S. aureus* reporter strain [27]. Isolation was performed by two-step activity-guided fractionation: a solid-phase extraction and HPLC purification. The active components have two UV maxima, the first at 227 nm and the second at 258–262 nm (Figure 1B). Mass spectrometric analysis in positive ion mode revealed three major ions at *m*/*z* 441.20, 527.20, and 541.22, corresponding to the [M+K]^+^ adducts of MLN A (C_24_H_34_O_5_), mal-MLN A (C_27_H_36_O_8_), and suc-MLN A (C_28_H_38_O_8_), respectively (Figure 1C–E and Appendix A). All the compounds exhibited a characteristic pattern of fragment ions at *m*/*z* 367.22, corresponding to the known MLN A core structure following the loss of two water molecules ([M-2H_2_O+H]^+^) and a secondary fragment at *m*/*z* 349.21, representing further dehydration ([M-3H_2_O+H]^+^). This fragmentation pattern has been previously described and is considered characteristic of the macrolactin scaffold [6].

To unambiguously confirm that the identified macrolactins belong to the MLN A subfamily, the structures of the identified MLN A, mal-MLN A, and suc-MLN A were validated with NMR. The chemical shifts for the macrolactin scaffold of the investigated molecules (Appendix A) are in line with previously published data for mal-MLN A [14]. The variability among the molecules is substantiated by the differing number of signals arising from the acyl substitutions, as well as the distinctive chemical shift values observed in the HSQC spectra (Appendix A) [28]. Based on these findings, along with the characteristic HMBC correlations (Appendix A), it was possible to unambiguously assign three distinct forms.

### 2.2. Bacterial Susceptibility and Cytotoxicity of the Macrolactins

To characterize the antimicrobial activity of the identified macrolactins in more detail, the susceptibility of a model panel of bacteria toward the macrolactins was estimated, along with their cytotoxicity (Figure 2A). Macrolactins were highly active against Gram-positive bacteria, including the common human pathogen *S. aureus* (MRSA) and opportunistic pathogens like *Enterococcus faecalis* and *Micrococcus luteus*. By contrast, macrolactins were inactive against Gram-negative bacteria, such as *Pseudomonas aeruginosa* and *Escherichia coli*, including hypersensitive *Escherichia coli* BW25113 Δ*tolC* and Δ*lptD* strains. That indicates that Gram-positive bacteria are the primary native bacterial targets for macrolactins. Acylated macrolactins were more active than MLN A; suc-MLN A was also more active than mal-MLN A. The estimated MICs against MRSA *S. aureus* GFP were 1 μg/mL, 0.25 μg/mL, and 0.1 μg/mL for MLN A, mal-MLN A, and suc-MLN A, respectively.

The direct practical application of macrolactins is complicated by their considerable cytotoxicity. The estimated IC_50_ values for human cell line HEK293T are 15 ± 2 μg/mL, 2 ± 0.3 μg/mL, and 0.3 ± 0.05 μg/mL for MLN A, mal-MLN A, and suc-MLN A, respectively (Figure 2A). The cytotoxicity of macrolactins generally correlates with their antimicrobial activity, and the therapeutic index (IC_50_/MIC) is rather low, reaching 8 for mal-MLN A and 3 for suc-MLN A. However, to create efficient semisynthetic agents for antimicrobial therapy, further structure–activity relationship analysis should be provided to obtain macrolactin-based analogs with improved selectivity toward bacteria.

### 2.3. Inhibition of Bacterial Protein Synthesis In Vitro by the Macrolactins

It was recently proposed that the MLN A mechanism of action is translation inhibition [23]. To elucidate this observation and estimate the efficacy of acylated macrolactins as translation inhibitors, they were assessed in a prokaryotic cell-free translation system. The tested compounds (MLN A, mal-MLN A, and suc-MLN) exhibited concentration-dependent inhibition of protein synthesis with estimated IC_50_ values of 1.5 ± 0.2, 0.8 ± 0.2, and 0.7 ± 0.1 μg/mL for MLN A, mal-MLN A, and suc-MLN A, respectively (Figure 2B). Hence, acylated macrolactins demonstrate more potent inhibitory activity on protein translation in the prokaryotic cell-free system, which is in line with their improved antimicrobial activity. Consequently, we speculate that, similar to MLN A, macrolactin analogs primarily act through the suppression of protein synthesis since (i) macrolactins demonstrate similar concentration ranges for protein translation inhibition and antimicrobial activity, and (ii) the antimicrobial activity of macrolactins and the efficacy of protein translation inhibition of macrolactin analogs correlate with each other.

### 2.4. Transformation of Acylated Macrolactins in Staphylococcus aureus Cells

We hypothesized that macrolactins may undergo enzyme-mediated biotransformation in living bacterial cells, which could influence their antimicrobial activity. To test this hypothesis experimentally, we incubated mal-MLN A and suc-MLN A in the presence of *S. aureus* cells and analyzed culture media using HPLC-UV and LC-MS/MS. In control experiments, only slow spontaneous hydrolysis was observed, with approximate half-lives of 35 ± 8 h and 70 ± 20 h for mal-MLN A and suc-MLN A, respectively (Figure 3A–D). In contrast, living *S. aureus* cells degraded both mal-MLN A and suc-MLN A much more rapidly. The biodegradation kinetics were 15 ± 4 and 13 ± 5 times higher than spontaneous hydrolysis for mal-MLN A and suc-MLN A, respectively (Figure 3C,D). Moreover, *S. aureus* cells showed a different pattern of biodegradation products compared to the spontaneous hydrolysis of acylated macrolactins (Figure 3A,B,E,F). Detailed analysis of chromatograms revealed that *S. aureus* cells produced additional degradation products distinct from those resulting from spontaneous hydrolysis. For both mal-MLN A (*m*/*z* 527.21 [M+K]^+^) and suc-MLN A (*m*/*z* 541.22 [M+K]^+^), incubation with *S. aureus* yielded additional peaks at the same *m*/*z* but with shifted retention time (Appendix A) and different UV spectral patterns (Appendix A). Analogously, in both cases, we observed a peak at *m*/*z* 441.20 ([M+K]^+^), corresponding to the hydrolyzed malonyl- or succinyl-MLN F (Appendix A). All of these ions showed MS/MS spectra identical to the respective initial macrolactins, consistent with rearranged isomers (Appendix A).

Biotransformation products were isolated, and the chemical structures of compounds were defined using NMR. The structure of the most abundant product of suc-MLN A transformation was determined as 7-*O*-succinyl-macrolactin F (suc-MLN F) (Appendix A, Appendix A). Accordingly, mal-MLN A was transformed into mal-MLN F, which was validated with mass and UV spectra (Appendix A). Minor components with identical molecular masses, UV spectra, and retention times were observed during the transformation of mal-MLN A and suc-MLN A and then determined by NMR as MLN F (Appendix A, Appendix A). Biotransformation of MLN A backbone into the MLN F backbone resulted in a dramatic 25- to 100-fold reduction in the antimicrobial activities of all the compounds. The measured MICs against *S. aureus* were 25 μg/mL, 25 μg/mL, and 6 μg/mL for MLN F, mal-MLN F, and suc-MLN F, corresponding to 25-, 100-, and 60-fold reductions, respectively.

### 2.5. MLN Biosynthetic Gene Cluster Is Highly Abundant in Bacillus velezensis and Bacillus amyloliquefaciens Species

The *B. velezensis* K-3618 strain was originally co-isolated from potato tubers of the Charoit variety in a natural biocontrol agent screening study [29] as an endophytic microorganism displaying cellulolytic, amylolytic, and weak nitrogen-fixing activity. In laboratory and field tests, *B. velezensis* K-3618 demonstrated high efficiency as a complex-action biofungicide, which not only provided protection of agricultural plants from diseases but also led to an increase in their productivity.

To reveal the biosynthetic potential of *B. velezensis* K-3618, we sequenced its genome. The assembly of the genome resulted in 23 contigs, with a total genome length of 3,864,279 base pairs, GC-content of 46.40%, N50 value of 471,177 bp, and N90 of 163,640 bp (GenBank NZ_JBNVTP000000000.1). The corresponding L50 and L90 values were 3 and 8, respectively, indicating that a relatively small number of large contigs represent the majority of the genome sequence. CheckM [30] analysis estimated the genome completeness at 99.63%, with no detectable contamination (0%). An antiSMASH [31] analysis of the complete genome sequence of *Bacillus* sp. K-3618 revealed a biosynthetic gene cluster (BGC), exhibiting high similarity to the previously described macrolactin-producing clusters (MiBiG [32] entries BGC0000181 [15,33,34] and BGC0001383 [35]). The identified cluster includes a complete set of core and tailoring enzymes required for macrolactin production (Figure 4).

To estimate the potential metabolic landscape of *B. velezensis* K-3618 and related strains, we performed comparative genomic analysis with 30 of the phylogenetically closest genomes (Appendix A). We first reconstructed the reference ML phylogeny based on core SNPs. Intriguingly, the distribution of species attributions does not follow a taxonomy-wise phylogenetic grouping (Figure 4). More precisely, genomes attributed to *B. velezensis* and *B. amyloliquefaciens* were mixed among the phylogenetic clades. The strain K-3618 formed a two-leaf clade with another soil-dwelling isolate, BS89, belonging to *B. velezensis*, as other representatives of the wider clade, allowing us to classify K-3618 in the *B. velezensis* species.

Next, we examined the distribution of known BGCs among the selected references using antiSMASH [36]. The strain K-3618 contained nine clusters responsible for the synthesis of bacillaene, bacillibactin, bacilysin, butirosin A, difficidin, fengycin, surfactin, plantazolicin, and macrolactin H (Figure 5; Appendix A). These clusters were also found in all 30 reference genomes, along with bacillothiazol A, detected in 22 genomes coupled with two singletons, namely, mersacidin and marthiapeptide A (Figure 5). We then mined for the core genes within the cluster of macrolactin H biosynthesis and found that all 10 core genes with 100% similarity and coverage could be identified in all genomes (Appendix A). Hence, the observed effects of the biological control of pathogenic bacteria via macrolactins potentiated with the conserved metabolic fingerprint of bacillaene, bacillibactin, and bacilysin should be strongly conserved in *B. velezensis* and closely related *B. amyloliquefaciens* strains. While *B. velezensis* and *B. amyloliquefaciens* are very close in their phylogeny, metabolic discrimination was proposed for *B. velezensis* based on its ability to produce macrolactin [34]. Macrolactin-producing *B. amyloliquefaciens* strains could be reannotated accordingly; however, most of them have another difficidin/oxydifficidin cluster. This difference may have a particular impact and impair their functionality as biocontrol and plant growth promoters.

## 3. Discussion

Bacterial susceptibility to macrolactins according to the literature data varies by orders of magnitude: the MIC of MLN A against *S. aureus* is in a range of 1–10 μg/mL [21,22], the MIC of suc-MLN A is lower than 0.25 μg/mL [21], and the value for mal-MLN A is higher than 120 μg/mL [14]. Our comparative study of three macrolactins indicates lower antibacterial activity for MLN A compared to its acylated analogs, as well as the higher activity of suc-MLN A compared with mal-MLN A, which may be associated with its greater stability. Statistical analysis of their direct effects on translation in a cell-free system revealed significant differences in the translation inhibition level between MLN A and its derivatives, Mal-MLN A and Suc-MLN A, at concentrations of 0.1 and 0.5 μg/mL (*p* < 0.001, *t*-test). In contrast, the differences between Mal-MLN A and Suc-MLN A were not statistically significant (*p* > 0.05, *t*-test). We observe that the differences in the activity of these macrolactins in bacterial cells are more significant than the differences in the inhibition of protein synthesis in a cell-free system. This may be due to the ability of acylated compounds to penetrate bacterial cells more effectively. A clear correlation between the minimum inhibitory concentration and toxicity was also observed, which complicates the prospects of using these compounds as antibiotic drugs. To create effective semisynthetic drugs for antimicrobial therapy, further analysis of the structure-activity relationship is necessary to obtain macrolactin-based analogs with improved selectivity to bacterial translation machinery.

In experiments involving incubation of antibiotics with bacterial cells, we observed two effects: dramatic acceleration of the degradation process by living *S. aureus* cells and the formation of new metabolites that were not detected after spontaneous hydrolysis of macrolactins. We found that biodegradation of mal-MLN A and suc-MLN A occurs through the rearrangement of the molecules with the formation of a keto group instead of a hydroxy group with a double bond reduction. Partial hydrolysis with the cleavage of the side acyl group also occurs. Both types of modified compounds, despite having similarity to the parent compound structure, are drastically less active.

Modification of molecules during biodegradation is a common pathway for antibiotic resistance in bacteria, including staphylococci, along with such processes as enzymatic modification of the binding site, drug efflux, reduced permeability, and metabolic adaptation [37]. The mechanisms of inactivation of macrolides include a variety of modifications, such as hydrolysis, reduction, acetylation, and deacylation [38]. Esterases EreA and EreB, found in *S. aureus*, cleave the macrocyclic ester linkage, providing resistance to the class of macrolide antibiotics. As shown in the example of tetracycline degradation, modification of an antibiotic can sometimes lead to selection against resistance due to the activity of stable degradation products [39].

We observed that macrolactins undergo biotransformation in *S. aureus* cells, resulting in MLN A backbone rearrangement that could be described as an oxidation of α,β-unsaturated alcohols to ketones, followed by double bond reduction leading to less toxic MLN F backbone products. Although enzymatic redox reactions of unsaturated compounds have been quite extensively studied, we have not found any examples of one-step biotransformation, such as the rearrangement detected for macrolactins. The possible pathways for two-step biotransformation may somehow simulate steps of fatty acid synthesis, including oppositely directed oxidation of α,β-unsaturated alcohols followed by double bond reduction. The closest examples of oxidation reactions involving bacterial enzymes were reported for whole-cell biocatalysis processes. Biocatalytic oxidation of racemic (hetero)aromatic sec-alcohols to the corresponding ketones has been reported for alcohol dehydrogenase-modified *E. coli* cells [40]. Enzymatic enantioselective oxidation of allyl alcohols has also been reported [41]. An example of biotransformation in Gram-positive pathogen cells is the hydroxylation and oxidation of thymol with the formation of a less toxic product [42]. Conversely, demethylation and oxidative deamination of sumatriptan in cells of various pathogens, including staphylococci, produce metabolites more toxic than the original drug [43]. After screening research, ten bacterial and fungal organisms capable of transforming cyclic and heterocyclic ketones into the corresponding alcohols, yielding pure complex chiral molecules, were revealed previously [44]. Hence, the discovered biotransformation of the MLN A backbone into the MLN F backbone is a specific, uncommon feature of macrolactins. However, similar antibiotic-inactivation mechanisms may be relevant to unsaturated macrolides or polyene antibiotics.

The biotransformation reaction of macrolactins is of particular interest, as it provides the molecular pathways for antibiotic inactivation and development of antibiotic resistance in pathogenic *S. aureus* strains, as well as natural inactivation of macrolactins. We believe the discovered biotransformations will bring new strategies for creating new antibiotics by stabilizing the architecture of a macrolactam backbone and introducing unnatural acyl analogs.

We revealed the molecular fingerprint of antibacterial metabolites of a common biocontrol agent, *Bacillus velezensis* K-3618. To identify the most active antibiotics, model live biosensors based on hypersensitive *S. aureus* and *E. coli* (Δ*tolC* and Δ*lptD*) strains were applied to ensure detection of even a trace of antimicrobial activity. Moreover, genomic characterization enabled prediction of all the main secondary metabolites of this strain. Direct antimicrobial activity-guided metabolomic analysis of *B. velezensis* growth culture demonstrated that the overwhelming majority of antimicrobial activity in the metabolome corresponds to macrolactins. Future research in plant-based models will be useful for understanding the ecological role of the production of macrolactins by *Bacillus* biocontrol agents. While our findings demonstrate strong antimicrobial potential, further validation using plant-pathogen models would be necessary to fully substantiate the agricultural relevance of these metabolites. However, we suggest that *B. velezensis*, in addition to direct biocontrol effects [2,7,8], may even have an indirect effect on phytopathogens by competing for nutrients, while the production of macrolactins provides selective pressure and promotes the shaping of the microbiome’s composition. The macrolactin’s BGC is highly conserved in *B. velezensis* and closely related *B. amyloliquefaciens* species, which stresses its high significance for the biology of these species. We speculate that the production of macrolactins, along with the conserved metabolic fingerprint of bacillaene, bacillibactin, and bacilysin, is essential for their functionality as efficient biocontrol agents. However, the functionality of the difficidin–oxydifficidin cluster may be a valuable hallmark of *B. velezensis*-based biocontrol agents. The abundance of the macrolactin’s BGC allows us to expand the obtained results to the numerous kinds of *B. velezensis* that are currently applied in agriculture, and we believe that the high activity of MLN A backbone analogs, combined with their natural biodegradation pathway, is of prime importance. Thus, macrolactins’ antimicrobial potential is constrained by biodegradation, but this also suggests evolutionary balancing of their ecological function.

## 4. Materials and Methods

### 4.1. Bacterial Strains and Cell Lines

The macrolactin-producing strain *Bacillus velezensis* K-3618 was obtained from the collection of the All-Russian Research Institute of Agricultural Microbiology, ID RCAM 07246 (https://en.arriam.ru/kollekciya-kul-tur1, accessed on 1 March 2024). The strain K 3618 was deposited as isolated from potato tubers of the Charoit-variety endophytic microorganism, with cellulolytic, amylolytic, and weak nitrogen-fixing activity. *Escherichia coli* ATCC 25922, *Micrococcus luteus* ATCC 4698, *S. aureus* ATCC 43300, *S. aureus* ATCC 25923, and *S. aureus* ATCC 29213 are from the American Type Culture Collection (ATCC). A bacterial collection including *Bacillus subtilis* 168, *Enterococcus faecalis* 125, *Enterococcus faecium* 40, *Enterococcus faecalis* 128, *Macrococcus caseolyticus* 107, *Escherichia coli* BL21 (DE3), *Pseudomonas aeruginosa* 51911, *Escherichia coli* BW25113 Δ*tolC*, and Δ*lptD* was kindly provided by Lytech Co., Ltd. (Moscow, Russia) [27]. *Staphylococcus aureus* GFP constitutively producing GFP was kindly provided by Andrey Shkoporov from the Department of Microbiology and Virology, Russian National Research Medical University, Moscow. Cell line HEK293T cells were kindly provided by Dr. E. Knyazhanskaya.

### 4.2. Genome Sequencing, Assembly, and Annotation

The Wizard Genomic DNA kit (Promega, USA, Madison, WI) was used to isolate genomic DNA. The DNA library was prepared using the KAPA HyperPlus kit (Roche, Switzerland, Basel). Whole-genome sequencing was performed at the Lopukhin Center for Genomics, Proteomics, Metabolomics (Russia) using the NextSeq 1000 platform (Illumina, USA, San Diego, CA) and NextSeq 1000/2000 P1 reagents (300 cycles).

Raw sequencing reads were initially subjected to quality assessment using FastQC (v0.12.1) [45], and subsequent filtering of low-quality and adapter-contaminated reads was performed with fastp (v0.23.2) [46]. The genome assembly was conducted using the SPAdes toolkit (v4.0.0) [47], followed by assessment of assembly metrics via QUAST (v5.2.0) [48]. To assess taxonomic placement and evaluate genome quality, we applied CheckM (v1.2.2) [30], and the closest related genomes were identified using fastANI (v1.33) based on comparisons with RefSeq [49] assemblies of Bacillus species from the NCBI database. Genome annotation was performed using Prokka (v1.14.6) [50].

### 4.3. Phylogeny Reconstruction

To reconstruct the reference phylogeny, we first obtained the 30 closest genomes relative to our strain according to ANI (Average Nucleotide Identity) estimates with all available genomes of the Bacillales order deposited in the RefSeq Assembly [49] database. The pair-wise genome similarity was evaluated with the fastANI v1.33 tool [51]. The selected dataset was then used to build a pangenome applying Panaroo v1.2.8 [52] launched in the “strict” mode. MAFFT aligner v.7 [53] was specified to align core genes defined as those present in no less than 95% of genomes. Based on the concatenated alignment of core genes, we determined the best-fit evolutionary model with ModelTest-NG v0.1.7 [54]. To reduce insignificant signals, we extracted SNPs (Single-Nucleotide Polymorphisms) with SNP-sites v2.5.1 [55]. The resulting alignment and selected evolutionary model were used for phylogeny reconstruction via the ML (Maximum Likelihood) approach implemented in the RAxML-NG v1.1.0 program [56]. A total of 1000 bootstrap replicates was specified. Visualization of the results, both the reference tree and the ordered data, such as BGCs, was performed by applying the ggplot2 v3.3.5 [57] and ape v5.6-2 [58] packages.

### 4.4. Cultivation, Isolation, and Purification of Macrolactins

The strain K-3818 was cultivated on the agar medium at a temperature in the range of 37 °C. The strain was transferred from the surface of the agar to a 750 mL Erlenmeyer flask with 50 mL of 2YT nutrient medium of the following composition, g/L: tryptone—16, yeast extract—10, NaCl—5, distilled water—up to 1 L, pH 7.0. Cultivation was carried out at 28 C for 24 h on a thermostat shaker Innova 40 (New Brunswick Scientific, USA, Enfield, CT) at 150 rpm. Using the first culture as an inoculum for seeding at a concentration of 3% *v*/*v*, the second-generation culture was grown in 150 mL on the same medium and under the same conditions.

Bacterial cells were eliminated from culture broth by centrifugation at 5000 rpm on a Sigma 3-16KL centrifuge and filtration through a 0.47 μm MCE membrane filter (Millipore, Burlington, MA, USA). In total, 1 L of cleared supernatant was loaded on a 30 mL cartridge packed with 7 g of LPS-500-H polymer sorbent (copolymer divinylbenzene–hydrophilic monomer, pore size 50–1000 Å, 70 μm, Technosorbent, Russia, Moscow) at a flow rate of 15 mL/min using a peristaltic pump (Masterflex L/S variable speed pump Systems, Masterflex, Germany, Gelsenkirchen). Extraction was performed using a syringe, passing 15 mL of a mixture of water and acetonitrile (ACN) at a rate of about 15 mL/min, successively, with ACN contents of 0, 10, 20, 35, 50, 75, and 100%. The activity of the collected fractions was tested using a reporter *S. aureus* GFP strain. Active fractions eluted at 35, 50, and 75% ACN were further analyzed by HPLC on an RP column using the Nexera X2 LC 30A instrument (Shimadzu, Japan, Kyoto), equipped with an SPD-M20A detector. HPLC conditions for the screening analysis were as follows: HC C18, 150 × 4.6 mm, 5 μm, 110 Å; eluent solvents, A–0.1% AA, B–MeCN; gradient elution, from 30% B to 90% B at 8 min, then 90% of solvent B; flow rate, 1.5 mL/min, UV 260 nm. HPLC fractions were collected and tested for activity, and the active fractions containing pure active substances were isolated and then analyzed by LCMS. Mal-MLN A was contained in fractions of 35 and 50% ACN; MLN A and suc-MLN A were in fractions of 50 and 75% ACN. All three fractions were combined for subsequent isolation of the compounds by HPLC on a semipreparative column—Gemini NX C18, 150 × 20 mm, 10 μm, 110 Å (Phenomenex, USA, Torrance, CA)—using the PuriFlash 5.250 instrument (Interchim, Montluçon, France). Isocratic elution at 43% of solvent B for 20 min at a flow rate of 16 mL/min was applied.

### 4.5. LC-MS Analysis

LCMS analysis was carried out on an Ultimate 3000 RSLCnano HPLC system connected to an Orbitrap Fusion Lumos mass spectrometer (ThermoFisher Scientific, Waltham, MA, USA), with the loading pump used for analytical flow gradient delivery. Samples were separated on a Gemini NX -C18 3 μm 100 Å column (Phenomenex, USA, Torrance, CA), 100 × 2.1 mm at a 200 μL/min flow rate in the linear gradient of acetonitrile in water, with the addition of 10 mM ammonium formate and 0.1% FA. UV data were collected at 275 nm. MS1 and MS2 spectra were recorded at 30K and 15K resolutions, respectively, with HCD fragmentation. Raw data were collected and processed on Thermo Xcalibur Qual ver. 4.3.73.11.

### 4.6. NMR

The structures of the compounds were elucidated using the conventional heteronuclear NMR approach. The compounds presumably identified according to HRMS as macrolactin A, 7-*O*-malonyl macrolactin A, and 7-*O*-succinyl macrolactin A were dissolved in CDCl3 (99.8%, Solvex, Russia, Moscow) and placed in a 5 mm NMR tube (Wilmad, Vineland, NJ, USA). Other samples were dissolved in MeOD-d4 (100%, Sigma-Aldrich, St. Louis, MO, USA) and placed in a 5 mm Shigemi NMR tube (SHIGEMI, Tokyo, Japan). NMR spectra were recorded at 30 °C using a Bruker Avance II 700 MHz NMR spectrometer (Bruker, Billerica, MA, USA) equipped with a TCI cryogenic probe. The 1D 1H, 13C, and 2D 1H,13C-HSQC and 1H, 13C-HMBC, DQF-COSY, and 2D TOCSY spectra were recorded for each sample.

### 4.7. Antibacterial Activity Assessment

Determination of the activity of the pure compounds was carried out using serial 2X-fold dilutions starting from final concentrations of 50 μg/mL. Individual colonies of bacteria from the plate with 2YT agar medium (10 g/L yeast extract; 16 g/L tryptone; 5 g/L NaCl; 15 g/L agar) were transferred to 5 mL of 2YT nutrient medium (10 g/L yeast extract; 16 g/L tryptone; 5 g/L NaCl) and incubated at 37 °C overnight. The overnight cultures were transferred to a fresh nutrient medium in a ratio of 1:100 and incubated at 37 °C for 3 h. Next, the cultures were diluted to OD600 ~ 0.001 a.u. and applied to the serial 2-fold dilutions of the compounds under investigation. MICs of antibiotics were determined after 16 h of incubation in a 96-well plate at a wavelength of 600 nm using a Varioskan multimodal reader (Thermo Fisher Scientific, USA, Waltham, MA).

### 4.8. Cytotoxicity Assessment

To prepare the MTT solution, thiazolyl blue tetrazolium bromide was dissolved in phosphate-buffered saline, pH = 7.4, to a concentration of 5 mg/mL, and sterilely filtered through a 0.2 μm filter. To prepare a solubilizing solution, 40% DMF was diluted in 2% acetic acid, and a total of 16% sodium dodecyl sulfate was added to the resulting solution; then, pH was adjusted to 4.7. Double dilutions of the test compounds were prepared in DMSO as a negative control in the culture medium. A final concentration series of eight 2-fold dilutions was started at 6.4 μg/mL. Incubation was carried out in a humidified atmosphere at 37 °C with 8% CO_2_ for 72 h. Cytotoxicity was assessed in HEK293T cells according to the method described in [59]. Cell viability was determined by colorimetric assessment of cell metabolic activity after incubation for 72 h. All measurements were performed in triplicate biological replicates. A total of 10 μL per well of MTT solution was added to the test samples to a final concentration of 0.45 mg/mL and incubated for an hour at 37 °C. Then, 100 μL of solubilizing solution was added to each well; the result was measured at 570 nm using a Varioskan multimodal reader (Thermo Scientific, USA, Waltham, MA).

A graph of the dependence of the average survival value on the concentration of the test substance plotted on a logarithmic scale was constructed using MS Excel 2013. A well with cells that did not contain the test compounds was taken as 100% viability. Relative survival was calculated as the ratio of the absorbance for a given well to the absorbance for a well without a drug, in percentages. The cytotoxic concentration for each compound (IC_50_) was defined as the concentration at which a cytotoxic effect was induced in 50% of the cells in the monolayer. Doxorubicin was used as the positive control.

### 4.9. Inhibition Assays in Cell-Free Translation System

To examine the inhibition of translation caused by MLN A, mal-MLN A, and suc-MLN, we used the *E. coli* S30 Extract System for Linear Templates (Promega, USA). The reaction mixture (total volume 5 μL) contained 2 μL of S30 Premix, 1.5 μL of S30 extract, 0.1 μL of the RNase inhibitor Ribolock, 0.5 μL of amino acid mixture, 0.5 μL of Fluc mRNA (200 ng/μL), and 0.5 μL of the tested compound (in 10% DMSO). Erythromycin (25 μg/mL) and 10% DMSO were used as controls. The mixture (without mRNA) was prepared on ice and incubated for 5 min at room temperature to allow the antibiotic to bind to the ribosome. After mRNA was added, translation proceeded for 20 min at 37 °C. After incubation, 5 μL of Bright-Glo (Promega, USA) was added. Luminescence was measured using a Victor X5 2030 plate reader (Perkin Elmer, USA, Norwalk, CT). The results were normalized on DMSO.

### 4.10. Biotransformation of suc-MLN A and mal-MLN A by Staphylococcus aureus

Macrolactins were solubilized in DMSO at a concentration of 5 mg/mL and added to 2YT medium containing 20 μg/mL chloramphenicol to achieve a final concentration of 0.25 mg/mL. *S. aureus* GFP was cultured in 2YT medium, supplemented with 20 μg/mL of chloramphenicol, up to a culture density of OD600 = 3. The bacterial cells were centrifuged at 5000× *g* for 10 min. The cell pellet was resuspended in 2YT medium supplemented with either mal-MLN A or suc-MLN A to attain a final concentration of 0.25 mg/mL and a bacterial cell density of 7.5 × 10^9^ CFU/mL. A strong inoculum effect was characteristic of macrolactins, decreasing their antimicrobial activity at high cell densities. Under the conditions of biotransformation, bacterial survival was more than 80%, which was determined by serial dilutions and plating. Two types of controls were prepared: 800 μL of the cell suspension without suc-MLN or mal-MLN A and 800 μL of each compound in 2YT media. In the small-scale experiment, there was only one sample of each kind (experimental suspension, control suspension, and control with a compound); in the large-scale experiment, there were 20 experimental aliquots of 800 µL. The resulting aliquots, as well as the control samples, were incubated at 37 °C with shaking at 200 RPM in conical 15 mL tubes. Aliquots of 50 μL suspension samples and control samples were taken after incubation periods of 1, 2, 4, 8, 16, and 24 h. In the large-scale experiment, after 24 h of incubation, all suspension samples were combined and prepared by the same procedure.

### 4.11. Analysis of Reaction Broth and Isolation and Purification of Target Compounds After Incubation of the Macrolactins with Staphylococcus aureus Cells

The 50 μL samples taken at fixed time points were centrifuged at 10,000× *g* for 10 min; thereafter, the supernatant and cell pellets were separately frozen in liquid nitrogen and stored at −80 °C. Prior to analysis, the medium samples were thawed and treated with the addition of 100 µL ethyl acetate and then vortexed for 15 s before incubation on a tabletop shaker for 5 min. They were then vortexed again for 15 s and centrifuged at 10,000× *g* for 5 min. The resulting ethyl acetate phase was separated and vacuum-dried. For further analysis, every dry residue sample was dissolved in 100 μL of a mixture of 0.1% acetic acid and acetonitrile (1:1) and analyzed by HPLC-UV or HPLC-MS/MS.

HPLC analysis was performed on an RP column using the Nexera X2 LC 30A instrument (Shimadzu, Japan) equipped with an SPD-M20A detector. HPLC conditions were as follows: Luna C18(2), 150 × 4 mm, 3 μm, 110 Å (Phenomenex) eluent solvent A–10 mM NH4OAc, pH 5, solvent B–MeCN; isocratic elution at 40% of solvent B for 15 min, gradient to 90% ACN for 17 min, and cleaning the column for 3 min; flow rate, 0.75 mL/min; UV, 260 nm. In the large-scale experiment, isolation and purification of the compounds of interest were performed on a semipreparative column, Gemini NX C18 150 × 10 mm, 5 μm, 110 Å (Phenomenex).

## 5. Conclusions

This study elucidates the molecular basis for the antimicrobial potential of the widely used biocontrol strain *Bacillus velezensis* K-3618 and establishes macrolactins as its principal effectors. Mechanistic assays indicate that MLN A-based macrolactins primarily act by inhibiting protein translation. Acylation at the 7-*O* position substantially enhances activity, with acylated analogs outperforming MLN A in prokaryotic cell-free systems at low micromolar concentrations. On the other hand, we discovered pathogen-mediated biotransformation of MLN A-based analogs into MLN F-based analogs, which reduces activity by approximately two orders of magnitude. These findings define two complementary design approaches for next-generation macrolactins—preserving and stabilizing the MLN A core while optimizing acylation to resist degradative conversion and to improve antimicrobial activity/selectivity. Future research directions may be focused on optimizing acylation and preserving the MLN A core. Our data suggest an ecological rationale for the observed degradability: such biotransformations may function as built-in attenuation, balancing efficacy with environmental persistence across *Bacillus* species in diverse niches. Together, these results provide a mechanistic and chemical blueprint for developing macrolactin-based antibacterials and improving the sustainability and reliability of *Bacillus*-driven organic farming.

## Figures and Tables

**Figure 1 ijms-26-11167-f001:**
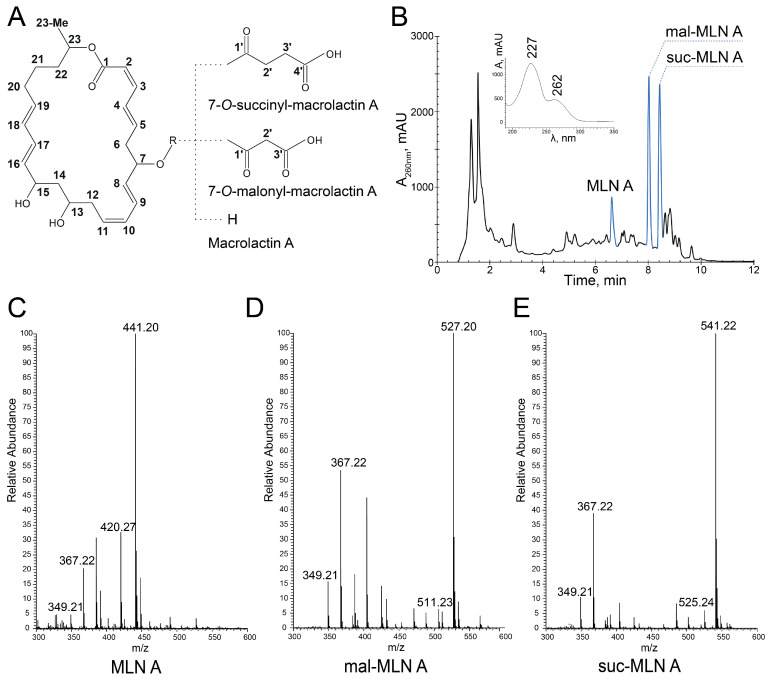
Isolation and identification of the macrolactins. (**A**) General structure of MLN A scaffold with R-group substitutions. (**B**) HPLC-UV of an active fraction extracted at 50% ACN, with the chromatogram recorded at 260 nm. The peaks corresponding to MLN A, mal-MLN A, and suc-MLN A are indicated in blue. Inset: UV absorption spectrum with characteristic maxima at 227 and 262 nm for the macrolactins. (**C**–**E**) Positive-ion mode HRESIMS of MLN A (**C**), mal-MLN A (**D**), and suc-MLN A (**E**).

**Figure 2 ijms-26-11167-f002:**
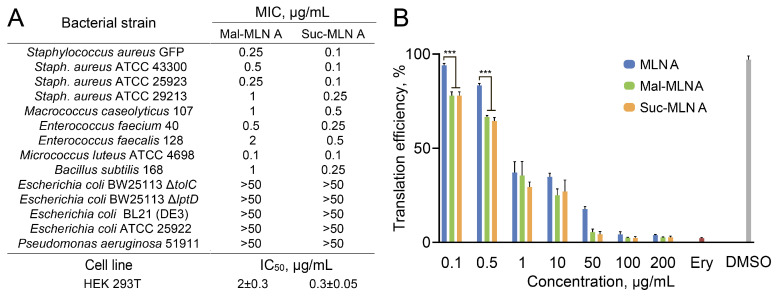
Antimicrobial activity of macrolactins is mediated by inhibition of protein translation. (**A**) Antimicrobial activity spectra and toxicity of the macrolactins. (**B**) Inhibition of protein synthesis in prokaryotic cell-free translation system by MLN A, mal-MLN A, and suc-MLN A. Erythromycin, 25 μg/mL, and DMSO were used as positive and negative controls, respectively. Data are represented as means of three biological replicates. Statistical significance between the derivatives and the parent MLN A compound at corresponding concentrations is indicated by asterisks: *** *p* < 0.001, *t*-test.

**Figure 3 ijms-26-11167-f003:**
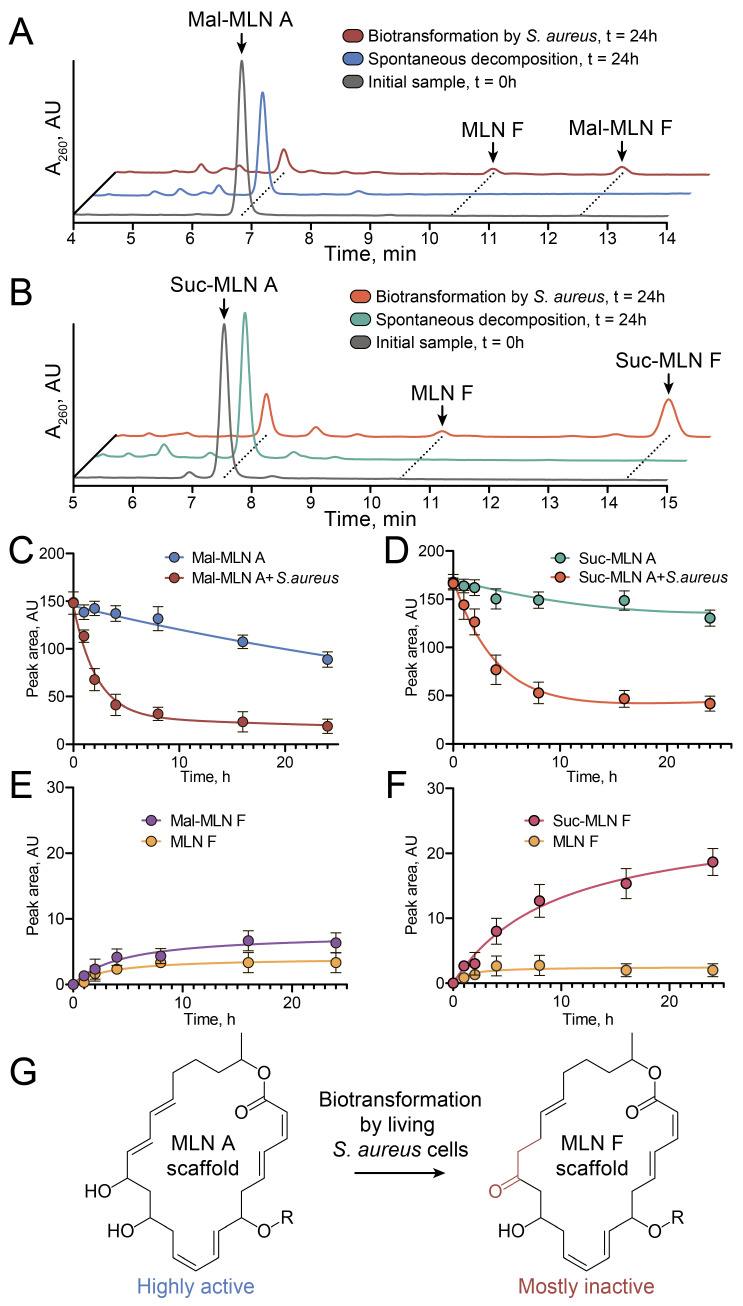
Biotransformation of the macrolactins incubated with *S. aureus* cells, according to data from HPLC analysis. (**A**,**B**) Chromatograms obtained at UV 260 nm for the samples of mal-MLN A (**A**) and suc-MLN A (**B**). Black curves correspond to the samples at the beginning of incubation with *S. aureus* cells (0 h); blue curves correspond to the samples incubated without *S. aureus* cells for 24 h; red curves correspond to the samples incubated with *S. aureus* cells for 24 h. (**C**–**F**) Kinetics of biotransformation according to the time dependence of peak areas in HPLC analysis. Mal-MLN A (**C**) and suc-MLN A (**D**) degradation over 24 h with bacterial cells and without them. Growth of mal-MLN A (**E**) and suc-MLN A (**F**) degradation products during incubation with bacterial cells. (**G**) Scheme of biotransformation of the macrolactins (for R-group substitution, see Figure 1A).

**Figure 4 ijms-26-11167-f004:**
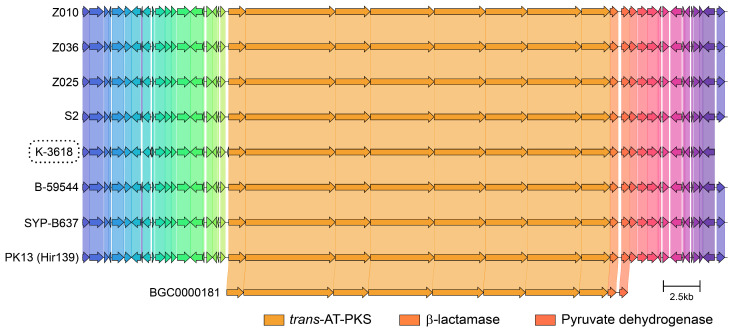
Comparison of the macrolactin BGCs (BGC0000181) with region 8.1 in the complete genome sequence of *Bacillus* sp. K-3618 and the selected homolog clusters in closely related species, generated using the clinker tool [31]. Homologous genes are color-coded.

**Figure 5 ijms-26-11167-f005:**
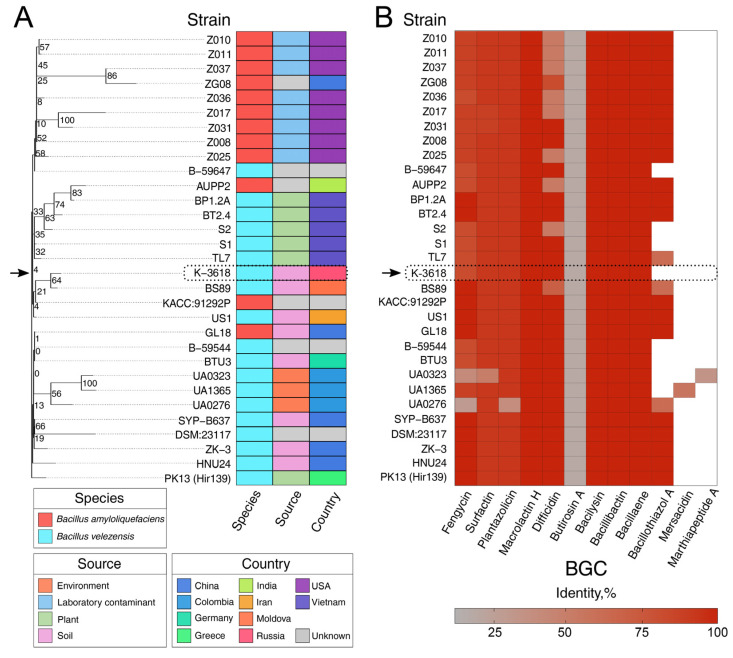
Phylogeny-wise distribution of BGCs (biosynthetic gene clusters) in the closest reference genomes relative to the *B. velezensis* K-3618 strain. (**A**) Reference ML (Maximum Likelihood) phylogeny reconstructed on core SNPs of 30 closest strains to the K-3618 strain based on the ANI (Average Nucleotide Identity) metrics. The tree was obtained using the RAxML-NG v1.1.0 software (see Materials and Methods). The numbers near the branches represent bootstrap support values with 1000 replicates. The adjacent left-most colored bars illustrate the strains’ metadata, namely, taxonomic attribution, isolation source, and country of origin for the studied strains, colorized according to their respective features. The black arrow points at the K-3618 strain. (**B**) The distribution of BGCs identified in the analyzed genomes using the antiSMASH v7.1 utility [36]. The rows correspond to strains ordered according to the reference phylogeny. The color of the tiles is proportional to the overall similarity with known BGCs calculated by the mentioned software.

## Data Availability

The raw genome sequencing data were submitted to the NCBI with BioSample number SAMN48451324, under BioProject PRJNA1262192. The assembled genome is available in the NCBI Assembly database under the accession JBNVTP000000000.

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
