# Peer review of "Bacillus-Based Biocontrol Agents Mediate Pathogen Killing by Biodegradable Antimicrobials from Macrolactin Family"

_ijms, 2025, doi:10.3390/ijms262211167_

Round 1

Reviewer 1 Report

Comments and Suggestions for Authors

In this paper, the metabolites of K3618 were isolated and identified, and the effect of its inhibitory protein was analyzed, which has certain scientific significance, but its mechanism of action has not been analyzed in detail. At the same time, there are still some minor problems in the article.
1、Line 132: I don't see TABLE S1. It is recommended that all tables and pictures can correspond.
2、Line 173: Drug-susceptible strains should be increased, including E. coli ATCC25922, etc.
3、Line 175, it should explain what drug sensitivity detection method is used. Is it a micro broth dilution method?
4、Line180:The estimated IC50 values for human cell line HEK293T are 15±2 μg/mL, 2±0.3 μg/mL, and 0.3±0.05 μg/mL for MLN A, mal-MLN A, and suc-MLN A, respectively . However, I think figure2 is wrong,2μg/ml and 0.3μg/ml?
5、Figure2B should add statistical analysis.

Author Response

In this paper, the metabolites of K3618 were isolated and identified, and the effect of its inhibitory protein was analyzed, which has certain scientific significance, but its mechanism of action has not been analyzed in detail. At the same time, there are still some minor problems in the article.

We gratefully thank the Reviewer1 for extremely valuable suggestions that allowed us to improve the MS considerably. Specifically,

1Line 132: I don't see TABLE S1. It is recommended that all tables and pictures can correspond.

We thank the Reviewer for carefully reviewing the supplementary materials and note that TABLE S1 is located on page 12 of the Supplementary Information materials file.

2Line 173: Drug-susceptible strains should be increased, including E. coli ATCC25922, etc.

We thank the Reviewer for his valuable suggestion to expand the number of macrolactin-susceptible strains tested. We additionally determined MIC values for the following strains, including E. coli ATCC 25922, S. aureus ATCC 43300, S. aureus ATCC 25923, S. aureus ATCC 29213, Enterococcus faecium 40, Enterococcus faecalis 128 Pseudomonas aeruginosa 51911. The obtained data have been added to the text of section 2.2. “Bacterial susceptibility and cytotoxicity of the macrolactins.” The respective MICs are provided in Fig. 2A

Lines 172-179

3Line 175, it should explain what drug sensitivity detection method is used. Is it a micro broth dilution method?

We thank the Reviewer for his question. Yes, we used a micro broth dilution method. Details of the methodology were specified in the Methods section, “4.7. Antibacterial Activity Assessment”

Lines 481-491

4.7. Antibacterial Activity Assessment

Determination of the activity of the pure compounds was carried out using the method of serial 2X-fold dilutions starting from the final concentrations of 50 μg/mL. Individual colonies of bacteria from the plate with 2YT agar medium (10 g/L yeast extract; 16 g/L tryptone; 5 g/L NaCl, 15 g/L agar) were transferred to 5 mL of 2YT nutrient medium (10 g/L yeast extract; 16 g/L tryptone; 5 g/L NaCl) and incubated at 37℃ overnight. The overnight cultures were transferred to a fresh nutrient medium in a ratio 1:100 and incubated at 37℃ for 3 h. Next, the cultures were diluted to the OD600 ~0.001 a.u. and applied to the serial 2-fold dilutions of compounds under investigation. MICs of antibiotics were determined after 16 h of incubation in a 96-well plate at a wavelength of 600 nm using a Varioskan multimodal reader (Thermo Fisher Scientific, USA).

4Line180The estimated IC50 values for human cell line HEK293T are 15±2 μg/mL, 2±0.3 μg/mL, and 0.3±0.05 μg/mL for MLN A, mal-MLN A, and suc-MLN A, respectively. However, I think figure2 is wrong, 2μg/ml and 0.3μg/mL.

We thank the Reviewer for his assistance in correcting errors and inaccuracies. The corresponding corrections in Fig. 2 have been made.

Lines 172-179

5Figure 2B should add statistical analysis.

In accordance with the Reviewer's comment, we performed a statistical analysis, the results of which are reflected in Fig. 2 and in the text of the Discussion section.

Lines 172-179

Figure 2B

Lines 178-179

Statistical significance between the derivatives and the parent MLN A compound at corresponding concentrations is indicated by asterisks *** p < 0.001, t-test.

Lines 312-316

Statistical analysis of their direct effect on translation in a cell-free system revealed significant differences in the translation inhibition level between MLN A and its derivatives, Mal-MLN A and Suc-MLN A, at concentrations of 0.1 and 0.5 μg/mL (p < 0.001, t-test). In contrast, the differences between Mal-MLN A and Suc-MLN A were not statistically significant (p > 0.05, t-test).

Reviewer 2 Report

Comments and Suggestions for Authors

The manuscript entitled ‘Bacillus-Based Biocontrol Agents Mediate Pathogen Killing by Biodegradable Antimicrobials from the Macrolactin Family’ presents an interesting and timely study that provides valuable insights into the antimicrobial potential of Bacillus-derived metabolites. The work is well-structured and supported by extensive experimental data. I believe the manuscript makes a meaningful contribution to the field; however, several aspects could be further refined to enhance clarity, readability, and overall impact.

Abstract

L19–23: The sentence is long; consider breaking it up for clarity (e.g., separate the rationale from the aim).

L26–33: Too much detail on macrolactin chemistry for the abstract. Suggest condensing to focus on the main findings: (i) identification of macrolactins as main antimicrobials, (ii) suc-MLN A is most active, (iii) biotransformation reduces activity.

L39–41: This ecological speculation is interesting but feels too detailed for the abstract—move it to the discussion.

Introduction

L54–59: This paragraph is a bit repetitive (“composition of secondary metabolites not characterized in detail”). Suggest merging sentences for conciseness.

L71–78: The macrolactin background is strong, but some structural detail (24-membered ring, olefinic bonds) may be better suited for Results/Discussion. Shorten here.

L120–127: Good summary of your study’s novelty. Strengthen by explicitly stating the “knowledge gap” (e.g., “Prior to this study, the specific active metabolites of B. velezensis K-3618 were unknown”).

Results

  • L131–141: Good start. Suggest clarifying why GFP-reporter S. aureus strain was used—important for readers unfamiliar with this assay.
  • L171–173: Provide statistical significance for MIC differences if available.
  • L179–185: Cytotoxicity results are important—add a comparison with therapeutic indices (e.g., MIC/IC50 ratio) to help readers interpret applicability.
  • L201–207: Clarify whether rapid degradation by S. aureus is enzyme-mediated (your discussion suggests it is, but here it reads as descriptive only).
  • L227–229: Quantify “dramatic reduction” in antimicrobial activity more explicitly (fold-change already given—highlight it as key finding).

Discussion

  • L307–315: The correlation between MIC and toxicity is well made—suggest adding a note about potential strategies to overcome toxicity (e.g., structural modification, formulation).
  • L316–323: Excellent description of degradation mechanisms. You could link more explicitly to clinical antibiotic resistance (e.g., macrolide-modifying enzymes).
  • L333–341: Consider citing additional examples of bacterial redox biotransformations to support plausibility.
  • L351–360: This section is strong—suggest finishing with a clearer “take-home” sentence (e.g., “Thus, macrolactins’ antimicrobial potential is constrained by biodegradation, but this also suggests evolutionary balancing of ecological function”).

Conclusion

L538–541: The two design approaches (preserve MLN A core + optimize acylation) are strong. Consider explicitly stating these as “future research directions.”

L541–543: The ecological rationale is interesting but could be shortened—focus on practical implications.

Author Response

The manuscript entitled ‘Bacillus-Based Biocontrol Agents Mediate Pathogen Killing by Biodegradable Antimicrobials from the Macrolactin Family’ presents an interesting and timely study that provides valuable insights into the antimicrobial potential of Bacillus-derived metabolites. The work is well-structured and supported by extensive experimental data. I believe the manuscript makes a meaningful contribution to the field; however, several aspects could be further refined to enhance clarity, readability, and overall impact.

We gratefully thank the Reviewer2 for extremely valuable suggestions that allowed us to improve the MS considerably. We are grateful to the Reviewer for a number of comments aimed at improving the clarity of the presentation of the material. We have revised the text in accordance with these recommendations. Specifically,

Abstract

L19–23: The sentence is long; consider breaking it up for clarity (e.g., separate the rationale from the aim).

We thank the Reviewer for the comment. We have corrected the text in accordance with this recommendation.

Lines 19-21

Rational use of organic farming approaches not only enables the reduction of costs and increased yields. It limits the risks associated with the use of pesticides and chemicals as well.

L26–33: Too much detail on macrolactin chemistry for the abstract. Suggest condensing to focus on the main findings: (i) identification of macrolactins as main antimicrobials, (ii) suc-MLN A is most active, (iii) biotransformation reduces activity.

We thank the Reviewer for the comment, we have corrected the text in accordance with this recommendation.

Lines 26-31

The identified macrolactin A (MLN A) and its acylated analogs 7-O-malonyl macrolactin A (mal-MLN A) and 7-O-succinyl macrolactin A (suc-MLN A) are active against Gram-positive bacterial pathogens, including multidrug-resistant strains. Among them, suc-MLN A is the most potent antimicrobial, highly active (MIC = 0.1 μg/mL) against the common human pathogen methicillin-resistant Staphylococcus aureus (MRSA).

Lines 34-36

We observed that acylated MLN A analogs undergo pathogen-mediated biotransformation to MLN F analogs, having the antimicrobial activity reduced by two orders of magnitude.

L39–41: This ecological speculation is interesting but feels too detailed for the abstract—move it to the discussion.

We thank the Reviewer for the comment, we have corrected the text in accordance with this recommendation to make it shorter.

Lines 38-40

However, we speculate that these degradability modes are of prime importance for bacterial ecology, and they are highly conserved in Bacillus species from various ecological niches.

Introduction

L54–59: This paragraph is a bit repetitive (“composition of secondary metabolites not characterized in detail”). Suggest merging sentences for conciseness.

We thank the Reviewer for the comment, we have corrected the text in accordance with this recommendation to make it shorter.

Lines 53-57

While a high diversity of bactericidal secondary metabolites of various chemical classes are known for Bacillus [4], the exact determining of the most principle-acting antimicrobials in biocontrol agents is of high interest since it enables a more comprehensive and targeted practical application of these strains [5,6].

L71–78: The macrolactin background is strong, but some structural detail (24-membered ring, olefinic bonds) may be better suited for Results/Discussion. Shorten here.

We thank the Reviewer for the comment, we have corrected the text in accordance with this recommendation to make it shorter.

Lines 69-74

Macrolactins (MLNs) are macrolide natural products from marine and terrestrial microorganisms, characterized by a 24-membered lactone ring [9,10]. First reported in 1989 as MLN A–F [11], the family had grown to 33 members by 2021 and continues to expand [12]. MLNs show broad pharmacological activity, most notably antibacterial effects. Structural diversity arises from variation in the number and placement of olefinic bonds in the ring and from different post-modifications.

L120–127: Good summary of your study’s novelty. Strengthen by explicitly stating the “knowledge gap” (e.g., “Prior to this study, the specific active metabolites of B. velezensis K-3618 were unknown”).

We thank the Reviewer for this suggestion. An additional sentence was provided to the introductory summary.

Lines 115-116

Prior to this study, the specific active metabolites of B. velezensis K-3618, as well as their mechanism of action and biodegradation pathways, were unknown.

Results

  • L131–141: Good start. Suggest clarifying why GFP-reporter S. aureus strain was used—important for readers unfamiliar with this assay.

We are grateful to the Reviewer for the comment. We add that the use of this strain is characterized by high sensitivity, which explains its use.

Lines 131-132

which was monitored using a highly sensitive alive biosensor based on the GFP-producing S. aureus reporter strain [27].

  • L171–173: Provide statistical significance for MIC differences if available.

We are grateful to the Reviewer for the comment. Statistical evaluation could be performed for translation inhibition studies. Such an evaluation was provided for Fig. 2 and the Discussion section.

Lines 172-179

Figure 2B

Lines 178-179

Statistical significance between the derivatives and the parent MLN A compound at corresponding concentrations is indicated by asterisks *** p < 0.001, t-test.

Lines 312-316

Statistical analysis of their direct effect on translation in a cell-free system revealed significant differences in the translation inhibition level between MLN A and its derivatives, Mal-MLN A and Suc-MLN A, at concentrations of 0.1 and 0.5 μg/mL (p < 0.001, t-test). In contrast, the differences between Mal-MLN A and Suc-MLN A were not statistically significant (p > 0.05, t-test).

  • L179–185: Cytotoxicity results are important—add a comparison with therapeutic indices (e.g., ratio) to help readers interpret applicability.

We are grateful to the reviewer for their comment, as the therapeutic index is a key indicator for drug substances. Typically, the value IC50/MIC is used for this purpose. We calculated the index for mal-MLN A  (8)  and for suc-MLN A  (3) and added them to the MS.

Lines 183-185

The cytotoxicity of macrolactins generally correlates with their antimicrobial activity, and the therapeutic index (IC50/MIC) is rather low, reaching 8 for mal-MLN A and 3 for suc-MLN A.

  • L201–207: Clarify whether rapid degradation by S. aureus is enzyme-mediated (your discussion suggests it is, but here it reads as descriptive only).

We thank the Reviewer for this suggestion. The additional mention of the enzymatic nature of the reaction suggested by the Reviewer has been included by us in the MS.

Lines 203-204

We hypothesized that macrolactins may undergo enzyme-mediated biotransformation in living bacterial cells, which could influence their antimicrobial activity.

  • L227–229: Quantify “dramatic reduction” in antimicrobial activity more explicitly (fold-change already given—highlight it as key finding).

We thank the Reviewer for this suggestion. The fold-change in antimicrobial activity was provided for all analogs.

Lines 229-232

Biotransformation of MLN A backbone to the MLN F backbone resulted in a dramatic 25- to 100-fold reduction of antimicrobial activities of all the compounds. The measured MICs against S. aureus were 25 μg/mL, 25 μg/mL, and 6 μg/mL for MLN F, mal-MLN F, and suc-MLN F, corresponding to 25-, 100-, and 60-fold reductions, respectively.

Discussion

  • L307–315: The correlation between MIC and toxicity is well made—suggest adding a note about potential strategies to overcome toxicity (e.g., structural modification, formulation).

We thank the Reviewer for this suggestion. The only reasonable strategy to overcome the cytotoxicity of macrolactins is a synthesis of new analogs specific to the procaryotic ribosome. Hence, the following sentence was added.

Lines 319-324

A clear correlation between the minimum inhibitory concentration and toxicity was also observed, which complicates the prospects of using these compounds as antibiotic drugs. To create effective semisynthetic drugs for antimicrobial therapy, further analysis of the structure-activity relationship is necessary to obtain macrolactin-based analogs with improved selectivity to bacterial translation machinery.

  • L316–323: Excellent description of degradation mechanisms. You could link more explicitly to clinical antibiotic resistance (e.g., macrolide-modifying enzymes).

We thank the Reviewer for this suggestion.  Resistance mechanisms are undoubtedly of great importance. Our data contribute to the understanding of these mechanisms. The mechanisms discovered are specific to macrolactins. However, it is highly likely that they could be found in microbiomes. We added the following sentence.

Lines 360-363

Hence, the discovered biotransformation of MLN A backbone to MLN F backbone is a specific uncommon feature of macrolactins. However, similar antibiotic-inactivation mechanisms may be relevant to unsaturated macrolides or polyene antibiotics.

  • L333–341: Consider citing additional examples of bacterial redox biotransformations to support plausibility.

We thank the Reviewer for this suggestion. We provide a number of additional references to publications to confirm the validity of our findings.

Lines 354-360

An example of biotransformation in Gram-positive pathogen cells is the hydroxylation and oxidation of thymol with the formation of a less toxic product [42]. Conversely, demethylation and oxidative deamination of sumatriptan in cells of various pathogens, including staphylococci, produce metabolites more toxic than the original drug [43]. After screening research, ten bacterial and fungal organisms capable of transforming cyclic and heterocyclic ketones into the corresponding alcohols, yielding pure complex chiral molecules, were revealed previously [44].

  • L351–360: This section is strong—suggest finishing with a clearer “take-home” sentence (“Thus, macrolactins’ antimicrobial potential is constrained by biodegradation, but this also suggests evolutionary balancing of ecological function”).

We thank the Reviewer for this suggestion. The Reviewer consideration is added

Lines 383-385

Thus, macrolactins’ antimicrobial potential is constrained by biodegradation, but this also suggests evolutionary balancing of ecological function.

Conclusion

L538–541: The two design approaches (preserve MLN A core + optimize acylation) are strong. Consider explicitly stating these as “future research directions.”

We thank the Reviewer for this suggestion. The following statement was added.

Lines 573-574

Future research directions may be focused on optimizing acylation and preserving the MLN A core.

L541–543: The ecological rationale is interesting but could be shortened—focus on practical implications.

We are grateful to the Reviewer for the suggested correction, but we believe the ecological significance of this result is also very important for Bacillus cell-cell interactions and microbiome integrity.

Reviewer 3 Report

Comments and Suggestions for Authors

   The introduction gives adequate background on Bacillus and macrolactin compounds, and the citations seem appropriate. Still, most of the text frames the topic around plant protection. The odd part is that the experiments don’t follow that direction—they use clinical strains, especially Staphylococcus aureus, rather than organisms relevant to crops. The same mismatch surfaces in the discussion, which makes it difficult to tell what the paper’s real focus is supposed to be. If the intent is agricultural impact, there needs to be some explanation for using human pathogens or, better, data from plant-related models.

   The overall approach—finding the active molecules, determining their structure, exploring biosynthetic genes, and checking how they work—is reasonable, but it is not aligned with the way the study is framed at the start. The description of the experimental steps is generally clear. The section on strains, extraction, metabolite analysis, and MIC testing is easy enough to follow.

   A major issue comes up in the transformation assays with S. aureus. The authors report MIC values of 0.25 and 0.1 μg/mL for mal-MLN A and suc-MLN A, yet in the assay they used 0.25 mg/mL. That is in the range of a thousand-fold higher than the inhibitory level. With concentrations that high, cells are likely to be damaged or dead, so the conversion they describe may not reflect active metabolism. It would need justification or additional tests using near-MIC levels and controls that show the cells are still viable.

   The figures and tables are readable and the main findings are easy to follow. What is missing is any evidence involving phytopathogens, which makes the references to crop applications hard to support. The conclusions about the compounds’ mode of action and chemical modification fit the data, but the claims about potential field use go beyond what the experiments actually cover.

Author Response

We gratefully thank the Reviewer3 for extremely valuable suggestions that allowed us to improve the MS considerably. We have revised the text in accordance with these recommendations. Specifically,

The introduction gives adequate background on Bacillus and macrolactin compounds, and the citations seem appropriate. Still, most of the text frames the topic around plant protection. The odd part is that the experiments don’t follow that direction—they use clinical strains, especially Staphylococcus aureus, rather than organisms relevant to crops. The same mismatch surfaces in the discussion, which makes it difficult to tell what the paper’s real focus is supposed to be. If the intent is agricultural impact, there needs to be some explanation for using human pathogens or, better, data from plant-related models.

We thank the Reviewer for this suggestion. The central question of the MS is what are the most active antibiotics produced by a widely applied biocontrol agent – Bacillus velezensis K-3618 strain? As it was mentioned in the Introduction section,

Lines 58-59

However, for the overwhelming majority of industrial biocontrol strains, the exact composition of principle-acting secondary metabolites was not characterized in detail.

Obviously, there is nothing new in the fact that Bacillus velezensis strains are widely applied biocontrol agents. What is new is how they mediate their antibacterial activity: what antibiotics they produce and how they could be biodegraded. Hence, the identification of the most active antimicrobials and their functional studies was the main focus of the MS.

The overall approach—finding the active molecules, determining their structure, exploring biosynthetic genes, and checking how they work—is reasonable, but it is not aligned with the way the study is framed at the start. The description of the experimental steps is generally clear. The section on strains, extraction, metabolite analysis, and MIC testing is easy enough to follow.

We thank the Reviewer for the valuable comment. Again, we see no discrepancy between the central question of the MS (identification of the main antimicrobials produced by Bacillus velezensis) and the methodology applied. To identify the most active antibiotics, model live biosensors based on hypersensitive S. aureus and E. colitolC and ΔlptD) strains were applied to ensure detection of even a trace of antimicrobial activity. Moreover, genomic characterization enabled prediction of all the main secondary metabolites of this strain. Direct antimicrobial activity-guided metabolomic analysis of Bacillus velezensis growth culture clearly showed that the overwhelming majority of antimicrobial activity in the metabolome corresponds to macrolactins. Hence, we suggest that the central question of the MS was clearly resolved.

A major issue comes up in the transformation assays with S. aureus. The authors report MIC values of 0.25 and 0.1 μg/mL for mal-MLN A and suc-MLN A, yet in the assay they used 0.25 mg/mL. That is in the range of a thousand-fold higher than the inhibitory level. With concentrations that high, cells are likely to be damaged or dead, so the conversion they describe may not reflect active metabolism. It would need justification or additional tests using near-MIC levels and controls that show the cells are still viable.

We thank the Reviewer for the valuable suggestion about the antibiotic’s concentration. We would like to make some clarifications on this issue. Macrolactins display a strong inoculum effect – MIC strongly depends on the concentration of cells in the growth culture. As the density of culture increases, the resistance of bacteria increases as well. As it was mentioned in the Materials and Methods section, dramatically different culture densities were used for determining MICs and biotransformation. For MICs, standard cell densities of OD600 ~0.001 a.u were used. This density was too low to produce reasonable quantities of macrolactins to determine their biodegradation products (MLN F-based). Hence, to increase production quantities, we were forced to scale up the process and increase cell density. For biotransformation experiments, the cell density corresponded to 7.5×109 CFU/mL, OD600 ~5 a.u. Accordingly, with a more than three-order-of-magnitude increase in density, the resistance of staphylococcal strains increases similarly. And at a concentration of 7.5×109 CFU/mL used for biotransformation, bacterial survival was more than 80%, which was determined by serial dilutions and plating.

Hence, we added the respective explanatory sentence to the Materials and Methods section:

Lines 532-535

A strong inoculum effect was characteristic for macrolactins, decreasing their antimicrobial activity at high cell densities. Under the conditions of biotransformation, the bacterial survival was more than 80%, which was determined by serial dilutions and plating.

The figures and tables are readable and the main findings are easy to follow. What is missing is any evidence involving phytopathogens, which makes the references to crop applications hard to support. The conclusions about the compounds’ mode of action and chemical modification fit the data, but the claims about potential field use go beyond what the experiments actually cover.

We thank the Reviewer for this insightful comment and for highlighting the discrepancy between the agricultural approach and the use of metabolites of B. velezensis strains against clinical pathogens. Meanwhile, the activity of these strains against various bacterial and fungal plant pathogens such as Ralstonia solanacearum, Fusarium oxysporum. Trichoderma aggressivum f. europaeum, Gaeumannomyces graminis var. tritici, Bipolaris sorokiniana, Xanthomonas citri subsp. Citri are widely known and were repeatedly experimentally established [2-8]. Macrolactins were found to be the compounds with antagonistic activity against A. tumefaciens C58 [5]. They also exhibited antifungal activity against F. oxysporum [6]. Taking into account the Reviewer’s suggestions, the activity spectrum of macrolactins, and the broad antagonistic activity of B. velezensis strains, we added that:

Lines 370-374

Future research should apply plant-based models to reveal the ecological role of the production of macrolactins by Bacillus biocontrol agents. However, we suggest that Bacillus may even have an indirect effect on phytopathogens by competing for nutrients, while the production of macrolactins provides the selective pressure and promotes shaping of microbiome composition.

Round 2

Reviewer 2 Report

Comments and Suggestions for Authors

The authors have revised manuscript carefully, therefore, i recommend it for publication.

Reviewer 3 Report

Comments and Suggestions for Authors

Thank you for the revisions and clarifications. The explanations about using S. aureus and E. coli as biosensors and the inoculum effect on antibiotic concentration are helpful. The added discussion on macrolactins’ ecological role and phytopathogen activity makes the study more coherent. Still, the agricultural focus remains a bit overstated, as no plant-based validation is shown. The connection between biocontrol relevance and clinical model choice could be better explained. Overall, the paper is clearer and better structured now, but a brief acknowledgment of these remaining limitations in the discussion would make it more balanced.